# Non-Invasive Predictive Biomarkers for Immune-Related Adverse Events Due to Immune Checkpoint Inhibitors

**DOI:** 10.3390/cancers16061225

**Published:** 2024-03-20

**Authors:** Ben Ponvilawan, Abdul Wali Khan, Janakiraman Subramanian, Dhruv Bansal

**Affiliations:** 1Department of Internal Medicine, University of Missouri–Kansas City School of Medicine, Kansas City, MO 64108, USA; ben.ponvilawan@umkc.edu (B.P.); abdulwalikhan@umkc.edu (A.W.K.); 2Inova Schar Cancer Institute, Fairfax, VA 22031, USA; janakiraman.subramanian@inova.org; 3St. Luke’s Cancer Institute, University of Missouri–Kansas City School of Medicine, Kansas City, MO 64108, USA

**Keywords:** immune checkpoint inhibitor, immunotherapy, immune-related adverse events, irAE, biomarker

## Abstract

**Simple Summary:**

Immune-related adverse events (irAEs) are one of the most common complications after cancer treatment caused by immune checkpoint inhibitors (ICIs). We performed an extensive review of the potential tests to predict irAEs in patients who receive ICIs. Approximately 40% of patients who receive ICIs experience irAEs. Despite this, only thyroid function tests are currently in mainstream use for predicting who will experience the adverse effects (namely, thyroid function abnormalities) from ICI treatment. A significant amount of research has paved the way for further potential tests that can be used in the clinic, but none have been meaningfully implemented in clinical practice. Nonetheless, further research on identifying these tests and incorporating them into clinical practice to help predict irAEs for patients with cancer will be significant in the field of ICI therapy.

**Abstract:**

Immune-related adverse events (irAEs) are the most common complication of immune checkpoint inhibitor (ICI) therapy. With the widespread use of ICIs in patients with solid tumors, up to 40% of the patients develop irAEs within five months of treatment, and 11% develop severe irAEs requiring interventions. A predictive test for irAEs would be a crucial tool for monitoring for complications during and after ICI therapy. We performed an extensive review of potential predictive biomarkers for irAEs in patients who received ICI therapy. Currently, only thyroid-stimulating hormone is utilized in common clinical practice. This is due to the unavailability of commercial tests and unclear predictive values from various studies. Given the lack of single strong predictive biomarkers, some novel approaches using composite scores using genomic, transcriptomics, cytokine levels, or clinical parameters appear appealing. Still, these have yet to be validated and incorporated into clinical practice. Further research conducted to validate the models before implementing them into real-world settings will be of the utmost importance for irAE prediction.

## 1. Introduction

Immune checkpoint inhibitors (ICIs) are one of the most important advancements in cancer treatment in the past ten years. They have shown efficacy in various types of solid tumors, including malignant melanoma, non-small cell lung cancer (NSCLC), and urothelial carcinoma [1,2,3]. Current ICIs are monoclonal antibodies that target programmed cell death protein 1 (PD1)/programmed death-ligand 1 (PD-L1), and cytotoxic T-lymphocyte associated protein 4 (CTLA-4) inhibitors, among others, such as lymphocyte-activation gene 3 (LAG-3), T-cell immunoglobulin and mucin domain-3 (TIM-3), and T-cell immunoreceptor with Ig and ITIM domain (TIGIT) inhibitors [4]. PD1, CTLA-4, LAG-3, TIM-3, and TIGIT are mainly T-cell surface proteins that serve as negative regulators for antitumor immunity, while PD-L1 is expressed on tumor cells serving as the ligand for PD1 [5]. The receptors and their ligand pairs are exhibited in Figure 1. These pathways are exploited by tumor cells to evade immunity, therefore sustaining their growth and forming a vital part of carcinogenesis. By inhibiting the immune checkpoints, the tumor cells are visible to the immune cells and can be eliminated.

However, inhibiting immune checkpoints can trigger the immunologic cascade through T-cell receptor diversity and form T-cell autoreactivity, resulting in a phenomenon called an immune-related adverse event (irAE) [6]. irAEs are rather common, occurring in almost 40% of patients within five months after ICI initiation, and 11% of patients had at least a grade 3 irAE, requiring ICI discontinuation or treatment for the irAE [7]. PD1/PD-L1 (PD(L)1) inhibitors are generally more tolerable, with notable irAEs including skin rash, thyroiditis, hepatitis, joint pain, pneumonitis, and neuropathy. CTLA-4 inhibitors are more associated with colitis, rash, hepatitis, and hypophysitis, while LAG-3 inhibitors are more often associated with colitis, hepatitis, rash, endocrinopathy, and neuropathy [5,8,9,10]. Table 1 summarizes the clinical abnormalities associated with irAE in each organ system.

Despite the known frequency and predilection of irAE by each ICI, they are often unpredictable in terms of onset and severity, resulting in the necessity of vigilance in those who received ICIs. With the increasing utilization of ICIs in patients with cancer, the need for predictive biomarkers for irAEs has been ever increasing. Here, we will discuss the existing evidence on biomarkers for irAEs based on an extensive literature review.

## 2. Biomarkers

A large number of studies have explored the potential predictive biomarkers for irAEs. These studies can be classified into two groups based on the association with the organ involvement of interest: organ-nonspecific and organ-specific.

### 2.1. Organ-Nonspecific

Due to the different mechanisms of action of anti-PD(L)1 and anti-CTLA-4, biomarkers vary between these two groups of regimens, as exhibited in Table 2.

#### 2.1.1. Anti-PD(L)1-Based Regimen

##### Autoantibodies

Autoantibodies are one of the most well-studied predictive biomarkers for irAEs. Several studies have shown that a pre-existing collection of autoantibodies such as antinuclear antibody, rheumatoid factor, or antithyroid antibodies (antithyroglobulin, and antithyroid peroxidase) before ICI treatment is associated with increased irAEs [11,12,13,14].

**Table 2 cancers-16-01225-t002:** Organ-nonspecific biomarkers for irAE.

Biomarker	Anti-PD(L)1-Based Regimen	Anti-CTLA-4 Monotherapy
Autoantibodies		
Any autoantibodies	Positive predictor [11,12,13,14]	Not predictive [15]
Anti-nuclear antibody	Conflicting result: Positive predictor [14,16,17,18,19] Not predictive [20,21,22,23,24,25,26,27]	
Rheumatoid factor	Conflicting result: Positive predictor [13] Not predictive [22]	
Raw cell counts		
Absolute lymphocyte count	Conflicting result: Positive predictor: ≥1500 [28], >2000 [29,30], >2600 [31], >820 2-week post treatment [32], >2000 1-month post treatment [30], no specific cutoff [33,34,35] Negative predictor: >1450 [36] Not predictive [12,37,38]	
Absolute neutrophil count	Conflicting result: Negative predictor: no specific cutoff [34,35,38] Not predictive [37]	Not predictive [39]
Absolute eosinophil count	Conflicting result: Positive predictor: >45 [40], >175 [41], no specific cutoff [30] Not predictive [12,42,43]	Not predictive [39]
White blood cell count	Conflicting result: Positive predictor: >27% increased from baseline [44] Negative predictor: no specific cutoff [34] Not predictive [33,35]	Not predictive [39,45]
Red blood cell count	Conflicting result: Positive predictor: no specific cutoff [34] Not predictive [35]	
Platelet count	Conflicting result: Positive predictor: >145,000 [31] Negative predictor: no specific cutoff [33,34] Not predictive [35,41]	
Cell type ratios		
Neutrophil–lymphocyte ratio	Conflicting result: Positive predictor: >2.3 [36], ≥4.3 [46], >6 [47] Negative predictor: >2.86 [48], ≥3 [42,49], ≥3.2 [33], ≥5 [50], >5.3 [31], no specific cutoff [34,38,51] Not predictive [29,32,35,37,40,46,52,53,54]	Not predictive [45]
Platelet–lymphocyte ratio	Conflicting result: Positive predictor: >165 [36] Negative predictor: ≥180 [54], >300 [52], >534 [31], no specific cutoff [38] Not predictive [29,33,35,40,51]	Not predictive [45]
Lymphocyte subset characteristics		
B cell	Conflicting result: Positive predictor: ≥30% reduction in total B cell count plus at least 2-fold increase in CD21 low-expressing cells or plasmablasts [55] Not predictive [41,56]	
T-cell subpopulation count	CD8^+^ T-cell count: reduced risk [41], not predictive [56] CD4^+^ T-cell, regulatory T-cell count: not predictive [41,56] CD8^+^CD38^+^Ki67^+^ T-cell expansion: positive predictor in NSCLC, but not in malignant melanoma [57] Ki67^+^ Treg-cell expansion: positive predictor in malignant melanoma, but not in NSCLC [57]	CD3^+^ and CD3^+^/CD4^+^ T-cell proportion: increased risk [58] Total T-cell count: not predictive [39]
T-cell diversity	T-cell receptor diversity [56] TCRB haplotype group 2: protective of grade ≥3 irAE [59]	T-cell receptor β-chain sequencing with ≥55 CD8^+^ T-cell clones or early increased number of T-cell clones [60,61]
Cytokines/Chemokines	Conflicting result: Positive predictor: IL6 [62], IL8, sLAG3, sPDL2, sHVEM, sCD137, sCD27, sICAM-1 [63], CXCL10 [57,64], CXCL13 [64], soluble CD163 (6s week post treatment) [65], CCL5 (4 weeks post treatment) [66] Negative predictor: IL10 [57], CXCL9, CXCL10, CXCL11, CCL19 [67] No correlation: IL6 [68], IL6, IL8, IL10 [37]	Positive predictor: IL17 [69], soluble CTLA-4 > 200 pg/mL [70] Negative predictor: IL6 [39], soluble MICA [71]
HLA	Positive predictor: HLA-B*35:01, HLA-B*40:02 [34], HLA-A3, HLA-DRB1*04:01 [29] Negative predictor: HLA-B*54:01 [34]	
SNPs	Positive predictor: *IFNW1* (rs10964859), *IFNL4* (rs12979860), *PD-L1* (rs4143815), and *CTLA-4* (rs3087243, rs11571302, and r27565213) [72] *GARBP*, *DSC2*, *BAZ2B*, *SEMA5A*, and *TYK2* SNPs [73] *TMEM162* (rs541169) [34] *MAPK1* (rs3810610), *ADAD1* (rs17388568) [74] Negative predictor: *UNG* (rs246079) [72] *miRNA-146a* (rs2910164) CC genotype [75,76] *PTPRC* (rs6428474), *IL6* (rs1800796) [74] *OSBPL6*, *AGPS*, *RGMA*, *ANKRD42*, *PACRG*, *FAR2*, *ROBO1*, *GLIS3*, *PVT1*, *PACRG*, *PREX2*, *HLA-DRB1*, *HLA-DQA1*, *HLA-DQA2*, and *TNFAIP3* SNPs [73] No association: *PDCD1* (rs2227981) [77]	Positive predictor: *SYK* (rs7036417) [78]
Other		
Frameshift neoantigen antibodies	Predictive of irAE [79]	
ctDNA alteration	*CEBPA*, *FGFR4*, *MET*, and *KMT2B* alterations: associated with irAE [80]	
RNA expression	Positive predictor: *LCP1* and *ADPGK* [56]	
PD-L1 expression	Positive predictor: PD-L1 expression ≥50% [36,81] Not predictive [37]	Negative predictor: PD-1 expression on CD4^+^ and CD8^+^ T-cells [58]
Mean platelet volume change	Negative predictor: decrease of ≤−0.2 fL [82]	
C-reactive protein	Positive predictor: >10 [83]	

Note: If not specifically denoted, the biomarker is a positive predictor for irAE. Abbreviation: ctDNA, circulating tumor DNA; CTLA-4, cytotoxic T-lymphocyte associated protein 4; HLA, human leukocyte antigen; irAE, immune-related adverse event; PD(L)1, programmed cell death protein 1/programmed death-ligand 1; SNP, single-nucleotide polymorphism.

However, the results were somewhat inconsistent when investigating which specific autoantibody was the main contributor to the association with irAEs. The most promising and well-studied autoantibody was anti-nuclear antibody [14,16,17,18,19,20,21,22,23,24,25,26,27], followed by rheumatoid factor [13,22], both studied in multiple types of solid tumors, including malignant melanoma and NSCLC. Some studies suggested a positive association between pre-existing autoantibodies and the development of any irAE [14,16,17,18,19], but some reported no association [20,21,22,23,24,25,26,27]. In summary, the evidence is conflicting, and they do not seem to be a reliable predictor for irAE in general when used alone.

##### Raw Cell Counts

Raw peripheral blood cell counts have also been extensively studied to predict irAEs. The absolute lymphocyte count was the most evaluated blood count among all studies. Although each study utilized different cutoffs (≥1500 [28], >2000 [29,30], and >2600 [31]), they favored the general trend of a higher pretreatment count as a positive predictor of irAEs [28,29,30,31,33,34,35]. An elevated absolute lymphocyte count during ICI treatment was associated with an increased risk of irAEs as well. An absolute lymphocyte count greater than 2000 at 1 month post ICI initiation was associated with grade ≥2 irAEs in patients with solid tumor (adjusted OR 1.81; 95% CI 1.03–3.25; *p* = 0.039) as reported by Diehl et al. [30]. Another study by Egami et al. reported the association of the absolute lymphocyte count greater than 820 at 2 weeks post ICI initiation with early onset irAEs (adjusted OR 3.58; 1.42–9.05; *p* = 0.007) in patients with NSCLC who received nivolumab monotherapy [32]. Despite this, few studies have reported no association [12,37,38], and one study observed a negative association [36]. Other white blood cell subsets include absolute neutrophil and eosinophil counts. The absolute neutrophil count has been reported to be negatively associated with irAEs as a continuous variable. However, no definite cutoff has been presented for predicting irAEs in these studies [28,29,31,32]. Two studies have shown a positive correlation between the absolute eosinophil count and irAEs. Ma et al. revealed that an absolute eosinophil count greater than 45 is associated with increased irAEs through anti-PD(L)1 therapy in patients with solid tumors [40]. A study by Bai et al. further supports an elevated likelihood of irAEs in patients who received both ICI-based combination therapy and ICI-monotherapy and had absolute eosinophil counts greater than 175 [41]. Another study by Diehl et al. reported that an elevated pretreatment and 1-month post-treatment absolute eosinophil count were both associated with grade ≥2 irAEs (OR (increase by 100/μL) 1.34; 95% CI 1.04–1.74; *p* = 0.027 and 1.21; 95% CI 1.02–1.51; *p* = 0.027, respectively) [30]. However, other studies could not confirm this association [12,42,43].

Total blood counts appear to be less understood and have yielded conflicting results. Fujisawa et al. reported that a relative white blood cell count increase of greater than 27% and a relative lymphocyte count decrease of greater than 23% were significantly correlated with grade ≥3 irAEs in patients with malignant melanoma who received nivolumab (OR 1.58; 95% CI 1.03–2.40 and 1.65; 95% CI 1.02–2.67, respectively) [44]. Another study by Michailidou et al. showed that a platelet count of >145,000 was associated with increased irAEs (adjusted OR 2.23; 95% CI 1.06–4.57; *p* = 0.03) [31]. Conversely, Sung et al. reported that white blood cell and platelet counts were negatively associated with irAEs. In contrast, the red blood cell count was associated with an increased risk of irAEs in patients with solid tumors. Gao et al. reported a negative association between irAEs and the platelet count, but not with the white blood cell count [33,34]. In addition, other studies have reported no association between any blood cell counts and irAEs [35,41].

##### Cell Type Ratios

Among the cell type ratios evaluated for irAE prediction, the neutrophil–lymphocyte ratio and platelet–lymphocyte ratio have been the most investigated in the literature. All studies suffered from heterogeneous cutoff values without a consensus on the optimal value for predicting irAEs. In addition, the results varied somewhat between studies. A majority of studies suggested that both ratios were negative predictors for irAEs (neutrophil–lymphocyte ratio > 2.86 [48], ≥3 [42,49], ≥3.2 [33], ≥5 [50], >5.3 [31], and no specific cutoff [34,38,51]; platelet–lymphocyte ratio ≥ 180 [54], >300 [52], >534 [31], and no specific cutoff [38]). However, several studies could not confirm the negative association of both ratios with irAEs [29,33,35,36,37,40,46,47,51,52,53,54]. Both raw cell counts and ratios from multiple studies suffered from the heterogeneity of the cutoffs, and the harmonization of the cutoff is necessary before this can be utilized in real-world settings.

##### Lymphocyte Subset Characteristics

Given the direct involvement of B and T lymphocytes in autoimmune processes, the divergence of lymphocyte characteristics from baseline was of interest in their association with irAEs [84,85]. Das et al. explored the differential composition of B cells in 39 patients with malignant melanoma who received anti-PD1, anti-CTLA4, or a combination of both. They observed that the concomitance of at least a 30% reduction in the total B cell count and at least a two-fold increase in CD21 low-expressing cells or plasmablasts were associated with grade ≥3 irAEs. No significant correlation was found between irAEs and other cell line counts, such as T- or NK cells [55]. However, another study by Bai et al. found that only a reduced CD8^+^ T lymphocyte count but not reduced B lymphocyte, CD4^+^, or regulatory T-cell counts were predictive of irAEs [41]. Jing et al. conducted a multi-omics study on 18,706 patients from multiple cancer types using the US Food and Drug Administration Adverse Event Reporting System (FAERS) and revealed no correlation between any B or T lymphocyte subset and irAEs. Instead, T-cell receptor diversity was positively correlated with irAEs [56]. Interestingly, a TCRβ variable gene polymorphism cluster obtained using the k-means clustering method exhibited a protective effect on the development of grade ≥3 irAEs [59]. Another study by Nuñez et al. showed differential positive predictive biomarkers for irAE: Ki67^+^ Treg cell expansion in malignant melanoma and CD8^+^CD38^+^Ki67^+^ T-cell expansion in NSCLC, while other cell populations were not predictive for the development of irAEs in both types of cancer [57]. This may signify different immunologic pathway activations that contribute to irAE in different cancers and explain a generally negative signal for T-cell subtypes as irAE biomarkers.

##### Cytokines and Chemokines

As one of the main drivers of inflammatory processes, cytokines, and chemokines were attractive choices as predictive biomarkers for irAEs [86]. A study by Tanaka et al. noticed that increased IL6 levels post treatment were found in all 13 patients with malignant melanoma who received nivolumab and had irAEs, while 5 out of 7 patients who did not have irAEs had decreased IL6 levels [62]. Another study with 79 patients with solid tumors was performed using a 34-cytokine/chemokine assay and revealed a positive association between increased IL8, sLAG3, sPDL2, sHVEM, sCD137, sCD27, and sICAM-1 levels and the incidence of irAEs [63].

Increased levels of CXCL10 and CXCL13 at baseline, but not of IL6 or IL8, were associated with grade ≥2 irAEs in patients with renal cell carcinoma [64]. Another study showed that an increase in CXCL10 and decrease in IL10 were associated with the development of an irAE in malignant melanoma and NSCLC [57]. However, two other studies further supported a lack of association between IL6, IL8, and IL10 levels and the development of irAEs [37,68].

Interestingly, Khan et al. investigated the levels of 40 cytokines and chemokines in patients with solid tumors who received ICIs. Lower pretreatment levels of CXCL9, CXCL10, CXCL11, and CCL19 were associated with irAEs in patients with solid tumors, with 6-week post-treatment levels of CXCL9 and CXCL10 that were significantly higher in those with irAEs than those without [67].

Cytokine and chemokine levels post treatment have also been explored, albeit to a lesser extent. Increased soluble CD163 levels 6 weeks post treatment were associated with increased irAEs in malignant melanoma patients (*p* = 0.0018) [65]. Another study on 34 cytokines in a cohort of 38 patients with NSCLC revealed that only increased CCL5 levels four weeks post treatment were associated with irAEs from nivolumab after multivariate adjustments (*p* = 0.0105) [66]. Again, cytokines/chemokine levels showed a variable predictive effect and do not appear to be a good predictor when used alone.

##### Human Leukocyte Antigen (HLA)

HLA variations and specific haplotypes are known to be associated with certain autoimmune diseases through deviations in antigen presentation and molecular mimicry [87]. Sung et al. evaluated HLA-B allelic variations in 672 patients with solid tumors who received ICIs and exhibited HLA-B*35:01 and HLA-B*40:02 as risk factors and HLA-B*54:01 as a protective factor for irAE development [34]. Another study by Abed et al. revealed that HLA-A3 and HLA-DRB1*04:01 are associated with increased risks of any irAEs [29]. With a limited number of studies evaluating HLA as biomarkers, further studies should be conducted on multiple patient populations to confirm the predictive role of each potential HLA biomarker.

##### Single-Nucleotide Polymorphisms (SNPs)

Many studies have focused on the SNPs of genes involved in immunologic pathways and their association with irAEs. The *PDCD1* (rs2227981) TT genotype has been shown to be associated with increased irAEs in an exploratory cohort of 181 patients with NSCLC who received nivolumab, but this was not reproducible in the validation cohort (OR 0.92; 95% CI 0.45–1.90; *p* = 0.828) [77]. An SNP of the *TMEM162* gene (rs541169) was noted to be a risk factor in a study on 672 patients who received ICIs [34]. *miR-146a*, an immunoregulatory microRNA with a role in CD4^+^ T-cell activation and neutrophil recruitment, was evaluated in two studies on 167 and 86 patients with solid tumors. The *miR-146a* (rs2910164) CC genotype was found to be associated with grade 3–4 irAEs [76]. 

Three studies utilized a custom large SNP panel for the identification of potential SNPs that had correlations with irAEs. Refae et al. utilized a 166 SNP panel on 94 patients with solid tumors who received anti-PD(L)1 therapy. *IFNW1* (rs10964859), *IFNL4* (rs12979860), *PD-L1* (rs4143815), and *CTLA-4* (rs3087243, rs11571302, and r27565213) SNPs were associated with increased grade ≥3 irAEs, while *UNG* (rs246079) SNP was associated with reduced irAEs [72]. A study by Xin et al. screened 340 patients for 37 SNPs and revealed increased irAEs in *MAPK1* (rs3810610) and *ADAD1* (rs17388568), and decreased irAEs were observed in *PTPRC* (rs6428474) and *IL6* (rs1800796) SNPs [74]. A multiplex genotyping in 89 patients with melanoma revealed 12 deleterious and 18 protective SNP variants for irAEs. The genes that were associated with the increased odds of irAEs included *GARBP*, *DSC2*, *BAZ2B*, *SEMA5A*, and *TYK2*, while those with decreased odds were *OSBPL6*, *AGPS*, *RGMA*, *ANKRD42*, *PACRG*, *FAR2*, *ROBO1*, *GLIS3*, *PVT1*, *PACRG*, *PREX2*, *HLA-DRB1*, *HLA-DQA1*, *HLA-DQA2*, and *TNFAIP3* [73]. Similar to HLA, further studies are needed to confirm the SNP effects in other populations.

##### Other Biomarkers

Further biomarkers apart from autoantibodies, cellular counts, and germline variants are still in the exploratory field for irAE prediction. Antibodies against pretreatment frameshift neoantigens were found to predict irAEs with 90% accuracy in lung cancer [79]. Circulating tumor DNA (ctDNA) alterations in *CEBPA*, *FGFR4*, *MET*, and *KMT2B* genes were positively associated with irAEs in 46 patients with gastric cancer who received anti-PD1 therapy [80]. An omics-based approach using DNA, RNA, and protein expressions from The Cancer Genome Atlas (TCGA) and FAERS databases identified pretreatment *LCP1* and *ADPGK* RNA expression levels to be positively correlated to the risk of irAEs. This was confirmed from a validation cohort of 28 patients with solid tumors, and it was more accurate than CD8^+^ T-cell abundance with T-cell receptor diversity for irAE prediction [56]. A high PD-L1 expression (≥50%) was found to be associated with increased irAEs in two studies [36,81]. However, another study by Hu et al. could not confirm this association [37]. Other potential biomarkers included a C-reactive protein >10 and decreased mean platelet volume change of ≤−0.2 fL, which were both associated with increased risk of irAEs [82,83].

#### 2.1.2. Anti-CTLA-4 Therapy Monotherapy

All studies investigating irAEs resulting from anti-CTLA-4 therapy were performed with patients with advanced or metastatic malignant melanoma. Autoantibodies, cell counts, and cell type ratios were not predictive of irAEs [15,39,45]. A study on 140 patients who received ipilimumab showed no correlation between irAEs and the absolute lymphocyte count or the total amount of a specific T-cell subpopulation (CD3^+^, CD4^+^, or CD8^+^) [39]. The further exploration of the effects of different distributions of pretreatment T-cell subgroups was performed by Damuzzo et al., who showed that patients with irAEs had higher CD3^+^ T-cell and CD3^+^/CD4^+^ T-cell populations. In addition, lower PD-1 expressions on CD4^+^ and CD8^+^ cells were associated with grade ≥3 irAEs [58]. T-cell receptor β-chain sequencing with ≥55 CD8^+^ T-cell clones or an early increased number of T-cell clones compared to pretreatment was associated with irAEs in prostate cancer patients [60,61].

For cytokines and chemokines, a comprehensive screening of cytokines and chemokines in 35 patients who received ipilimumab for neoadjuvant treatment before surgical resection revealed that only a higher pretreatment serum IL-17 level was associated with grade ≥3 irAEs [69]. In another study by Pistillo et al., soluble serum CTLA-4 (sCTLA-4) level of >200 pg/mL was associated with irAEs (OR 3.63; 95% CI 1.14–11.5, *p* = 0.029) [70]. On the other hand, a lower IL6 level was associated with grade ≥3 irAE, and a decreased soluble major histocompatibility complex class I chain-related protein A (sMICA) level was found in patients with irAEs (median level 69 vs. 130; *p* = 0.049) [39,71]. Interestingly, 42 SNP variants focused on germline immunomodulatory expression quantitative trait loci were evaluated by Ferguson et al. and revealed a *SYK* SNP (rs7036417) that was a positive risk factor for irAEs [78].

### 2.2. Organ-Specific

Given a predilection for organ systems that can be affected by ICI therapy, a large number of biomarkers have been investigated to predict irAEs specific for each organ system. Some organ-specific irAEs can be life-threatening, such as pneumonitis or hepatitis. Therefore, biomarkers that can predict irAEs for each organ system would be of great benefit for detecting symptoms in patients who have a higher risk of developing a particular irAE. Table 3 summarizes the biomarkers for organ-specific irAEs.

#### 2.2.1. Endocrine System

Within the endocrine system, the common manifestations of irAEs include primary thyroid dysfunction and hypophysitis. Rarer manifestations such as type 1 diabetes mellitus and adrenal insufficiency can also result from ICIs [88]. A study on 45 Japanese patients with advanced, unresectable malignant melanoma who received PD1 inhibitor therapy showed that a baseline absolute eosinophil count of >240/μL and relative eosinophil count at one month post treatment of >3.2% were associated with endocrine irAEs (OR 7.0; 95% CI 1.50–32.72; *p* = 0.013 and OR 5.11; 95% CI 1.23–21.28; *p* = 0.025, respectively) [43]. 

**Table 3 cancers-16-01225-t003:** Organ-specific biomarkers for irAE.

Organ System	Biomarker	Association
Endocrinologic		
Any	Absolute eosinophil count >240	↑ risk [43]
	Relative eosinophil count >3.2%	↑ risk [43]
	HLA-B*35:01	↑ risk [34]
Thyroid	Any autoantibodies	↑ risk for hypo/hyperthyroid but not hypothyroid alone [11]
	Antithyroid antibodies (antithyroid peroxidase or antithyroglobulin)	*Positive seroconversion associated with thyroiditis* [15] Pre-existing antibody associated with thyroid dysfunction [14,26,89,90,91,92,93]
	Antithyroid peroxidase	Pre-existing antibody [23,94,95,96] and post-treatment antibody [97] associated with thyroid dysfunction
	Antithyroglobulin	Pre-existing antibody [23,94,95,98,99] and post-treatment antibody [96,97] associated with thyroid dysfunction
	Thyroid-stimulating hormone	Elevated level: ↑ risk [73,90,92,96,98,99,100,101] Reduced level: associated with hyperthyroidism [98]
	Pretreatment IL1β, IL2, and GM-CSF	Elevated levels: ↑ risk [97]
	4-week post-treatment G-CSF, IL8, and MCP-1 levels	Reduced levels: ↑ risk [97]
	Neutrophil-lymphocyte ratio	Reduced level: ↑ risk [100]
	HLA-B*35:01	↑ risk [34]
Type 1 diabetes mellitus	HLA-C*01:02, HLA-DPA1*02:02, and HLA-DPB1*05:01, HLA-DQB1*04:01, HLA-DRB1*04:05, and HLA-DR4	↑ risk [102,103]
	Germline *NLRC5* and *CEMIP2* mutations	↑ risk [104]
Adrenocorticotropic hormone deficiency	Anti-pituitary antibody	↑ risk [105]
	HLA-Cw12, HLA-DR15, HLA-DQ7, and HLA-DPw9	↑ risk [105]
Hypophysitis	Anti-GNAL and anti-TIM2B	↑ risk [106]
	Anti-pituitary antibody	↑ risk [105]
	HLA-Cw12 and HLA-DR15	↑ risk [105]
	HLA-DQB1*06:02 and HLA-DRB4*01:01	↑ risk [107]
	HLA-DRB4*01:03	↓ risk [107]
Pituitary	HLA-B52, HLA-Cw12, HLA-DRB1*15:02, and HLA-DR15	↑ risk [108]
Dermatologic		
Any	Antinuclear antibody	Conflicting: No association [14,15] ↑ risk [19]
	Rheumatoid factor	↑ risk [14]
	Platelet count and neutrophil–lymphocyte ratio	Conflicting: No association [43] Elevated level: ↓ risk [100]
	Baseline plasma Ang-1 and CD40L	Elevated level: ↑ risk [109]
	Anti-BP180 IgG	↑ risk [110]
	*MAPK1* SNP (rs3810610)	↑ risk [74]
Vitiligo	White blood cell count	↑ risk [43]
Pruritus	HLA-DRB1*11:01	↑ risk [111]
Gastrointestinal		
Any	Elevated relative white blood cell and lymphocyte counts	↑ risk [44]
	HLA-DQB1*03:01	↓ risk [29]
	HLA-B*35:01	↑ risk [34]
Colitis	Elevated CD4^+^ T-cell count and reduced Treg cell %	*↑ risk* [112]
	Elevated IL6, IL8, and sCD25	*↓ risk* [112]
	Elevated IL17 and sCTLA-4 > 200 pg/mL	*↑ risk* [69,70]
	Overexpression of *CD177*, *CEACAM1*, *IGHA1*, *IGHA2*, *IGHG1*, and *IGHV4-31* at 3 weeks post treatment	*↑ risk* [113]
	Antinuclear antibody	↑ risk [25]
	White blood cell and absolute neutrophil counts	↑ risk [114]
	HLA-DQB1*03:01	↑ risk [111]
	HLA-A homozygosity	↑ risk [114]
Hepatitis	Neutrophil	Elevated level: ↓ risk [100]
	White blood cell count	↑ risk [114]
	Autoimmune hepatitis autoantibodies	No association [115]
Pancreatitis	White blood cell and absolute neutrophil counts	↑ risk [114]
	HLA-A homozygosity	↑ risk [114]
	*SMAD3* small sequence variations	↑ risk [114]
Hyperbilirubinemia	HLA-A*26:01	↑ risk [116]
Rheumatologic		
Any	HLA-DRB1*15:01	↓ risk [29]
	HLA-B*35:01	↑ risk [34]
	Rheumatoid factor	Pre-existing antibody: ↑ risk [117]
Inflammatory arthritis	HLA-DRB1*04:05	↑ risk [118]
	Rheumatoid factor and anti-cyclic citrullinated peptide	*No association* [15] ↓ risk [118]
Pulmonary		
Pneumonitis	Absolute eosinophil count ≥125	↑ risk [119]
	White blood cell count	↑ risk [120]
	Absolute neutrophil count	↓ risk [120]
	Neutrophil–lymphocyte ratio	No association [120], ↓ risk [100]
	Low IFN-γ	↑ risk [121]
	IL17 and anti-CD74 levels	Conflicting results: ↑ risk [106,109] Not associated [64]
	HLA-B35 and HLA-DRB1*11 haplotype	↑ risk [122]
	HLA-B*35:01	↑ risk [34]
	Anti-GAD	Pre-existing antibody associated with interstitial pneumonitis [23]
Neurologic		
Any	MCP-1 and BDNF levels	↑ risk [123]
Autoimmune encephalitis	HLA-B*27:05	↑ risk [124]
	Brain-reactive autoantibodies	No association [125]
Neuromuscular disease	Neuromuscular autoantibodies	↑ risk [125]
Cardiac		
Myocarditis	Absolute lymphocyte count	↓ risk [126]
	Neutrophil–lymphocyte ratio	↑ risk [126]
Hematologic		
Thrombocytopenia	HLA-DRB3*01:01	↑ risk [116]
Anemia and leukopenia	HLA-DPB1*04:02	↑ risk [116]
Other		
Vogt–Koyanagi–Harada-like uveitis	HLA-DRB1*04:05	↑ risk [127]

Texts in *italics and underlined* are factors that were studied with anti-CTLA-4 monotherapy. ↑ and ↓ represent “increased” and “decreased”, respectively. Abbreviation: CTLA-4, cytotoxic T-lymphocyte associated protein 4; HLA, human leukocyte antigen; irAE, immune-related adverse event; SNP, single-nucleotide polymorphism also reported to increase any endocrine irAE by Sung et al. in patients with solid tumors who received ICI therapy [34].

##### Thyroid Dysfunction

For thyroid irAEs, there has been a considerable amount of evidence of the association between the positivity of autoantibodies and the development of thyroid diseases.

Daban et al. explored autoantibodies in patients with solid tumors and revealed a positive relationship between pre-existing autoantibodies (antinuclear antibody, rheumatoid factor, antineutrophilic cytoplasmic antibodies, and antithyroid antibodies) and an increased risk for thyroid dysfunction (hypothyroidism or hyperthyroidism), but not hypothyroidism alone [11]. Positive pre-existing antithyroid antibodies appear to be a consistent risk factor. In seven studies, the presence of pre-ICI treatment antithyroid antibodies (either antithyroglobulin or antithyroid peroxidase) was associated with thyroid dysfunction [14,26,89,90,91,92,93]. Several studies further investigated antithyroid antibodies and revealed that the levels of both antithyroid antibodies in pretreatment and post-treatment settings were associated with an increased risk of thyroid dysfunction [23,94,95,96,97,98,99]. Interestingly, a cohort of patients with NSCLC who received pembrolizumab and developed hypothyroid irAEs exhibited a common thyroid function trajectory of initial transient hyperthyroidism within 1–2 months after pembrolizumab therapy, followed by prolonged hypothyroidism [93].

Baseline thyroid-stimulating hormone (TSH) levels might be a convenient alternative for predicting thyroid dysfunction from ICIs. Eight studies reported that an elevated baseline TSH level was associated with hypothyroid irAE from PD1 inhibitor-based therapy in solid tumors [73,90,92,96,98,99,100,101]. Conversely, a study by Wu et al. showed that a decreased level of TSH was predictive of hyperthyroidism [98]. However, no differences in baseline TSH levels were appreciated between patients with and without thyroid irAEs in another study by Maekura et al. [91].

Regarding cytokines and chemokines, a study on 18 cytokines before and four weeks after PD1 inhibitor-based therapy with patients with solid tumors showed that elevated pretreatment levels of IL1β, IL2, and GM-CSF were enriched in patients with thyroid dysfunction compared to those who did not. Reductions in G-CSF, IL8, and MCP-1 levels four weeks post treatment were also associated with thyroid irAEs [97]. Other potential biomarkers for thyroid irAE prediction include a decreased neutrophil–lymphocyte ratio and the presence of HLA-B*35:01 [34,100].

##### Other Endocrinologic Dysfunctions

Several HLA alleles and germline mutations were found to be risk factors for patients who developed type 1 diabetes after ICI therapy. In a cohort of 27 patients, the prevalence of the HLA-DR4 allele was significantly higher than that of the general population (76% vs. 17%, *p* < 0.0001). Interestingly, no differences were found in HLA-DR3, a known risk factor of type 1 diabetes [102]. Another study on 871 patients noted significantly increased frequencies of HLA-C*01:02, HLA-DPA1*02:02, HLA-DPB1*05:01, HLA-DQB1*04:01, and HLA-DRB1*04:05 alleles in patients with type 1 diabetes irAE (all *p* < 0.05) [103]. In another study by Caulfield et al., germline *NLRC5* and *CEMIP2* mutations were associated with an increased risk of developing type 1 diabetes post ICI therapy [104].

Similarly, a case-control study on 62 patients showed that HLA-Cw12, HLA-DR15, HLA-DQ7, and HLA-DPw9 alleles and the presence of anti-pituitary antibody pretreatment were risk factors of adrenocorticotropic hormone deficiency [105]. Meanwhile, HLA-Cw12, HLA-DR15, HLA-DQB1*06:02, and HLA-DRB4*01:01 alleles and anti-pituitary antibodies were also risk factors, and based on two studies by Kobayashi et al. and Quandt et al., HLA-DRB4*01:03 could be a protective factor for the development of hypophysitis [105,107]. Interestingly, Tahir et al. evaluated another approach to identify a novel biomarker using the combined RNA and protein expressions of autoantibodies in patients with solid tumors who received anti-PD1 with or without anti-CTLA-4 and developed hypophysitis irAEs. They showed that the increases in anti-guanine nucleotide-binding protein G subunit alpha (GNAL) and anti-integral membrane protein 2B (ITM2B) were associated with hypophysitis irAE in both discovery and confirmation cohorts [106].

A study by Yano et al. revealed enrichments of HLA-B52, HLA-Cw12, and HLA-DRB1*1502 alleles in patients with pituitary dysfunction from ICI therapy. Those with HLA-DR15 also had a predilection for a pituitary irAE compared to other endocrine dysfunctions such as type 1 diabetes, thyroid dysfunction, and pancreatitis [108].

##### Endocrine irAE from Anti-CTLA-4 Monotherapy

Of note, there has been limited evidence on endocrine irAEs through anti-CTLA-4 monotherapy. One study by de Moel et al. revealed an association between a positive seroconversion of any autoantibody before and after ICI treatment and the development of thyroiditis [15]. Another study by Pistillo et al. reported a non-significant trend toward increased irAEs and sCTLA-4 levels greater than 200 pg/mL in patients with malignant melanoma (OR 3.26; 95% CI 0.68–15.6, *p* = 0.14) [70].

#### 2.2.2. Dermatologic

A study on 137 patients with NSCLC who received PD1 inhibitors found that only pre-existing rheumatoid factor, but not antinuclear antibody, was significantly enriched in patients with dermatologic irAEs [14]. This observation was also noted in another study by de Moel et al. [15]. However, another study by Zhang et al. with 159 patients with NSCLC who received ICI therapy showed a positive relationship between pre-existing antinuclear antibody and increased risk of dermatologic irAEs [19].

A cohort of 10,344 patients from 15 phase II and III clinical trials utilizing atezolizumab showed that the platelet count and neutrophil–lymphocyte ratio were negatively associated with the risk of developing dermatologic irAEs [100]. However, Nakamura et al. could not confirm this association, while they did observe that a higher white blood cell count was a protective factor against vitiligo after multivariate adjustment with OR (increase by 100/μL) of 0.82; *p* = 0.0023 [43].

In another study by Tyan et al. with 52 patients who received ICIs, higher baseline plasma Ang-1 and CD40L levels were significantly associated with dermatologic irAEs [109]. Anti-BP180 IgG was associated with dermatologic irAEs, but not all irAEs, from anti-PD1/PD-L1 in NSCLC [110]. Other potential biomarkers include the *MAPK1* SNP (rs3810610), which was found to be a risk factor for dermatologic irAEs from a screening of 39 SNPs in 340 patients with solid tumors [74]. Another study by Ali et al. established an association between the HLA-DRB1*11:01 allele with pruritus from ICIs in malignant melanoma and NSCLC (OR 4.53, *p* = 0.0021) [111].

#### 2.2.3. Gastrointestinal (GI)

GI adverse events are one of the most common manifestations of irAEs. The most common manifestations include colitis and, to a lesser extent, hepatitis. Of note, it is well-known that treatment with anti-CTLA-4 has a higher incidence of GI irAEs compared to anti-PD(L)1 therapy [128,129].

A relative white blood cell increase of more than 59.1% and a relative lymphocyte count decrease of more than 32.3% were shown to have a significant correlation with lung or GI irAEs in patients with malignant melanoma who received nivolumab (OR 6.04; 95% CI 2.10–19.28 and 5.01; 95% CI 1.78–14.17, respectively). In this study, relative changes in the neutrophil, monocyte, or eosinophil counts did not correlate with irAEs [44]. HLA-B*35:01 was found to be a risk factor for any GI irAE [34]. On the contrary, HLA-DQB1*03:01 was a protective factor in patients with NSCLC who received anti-PD(L)1 monotherapy [29].

##### Colitis

A large body of evidence has reported several risk factors or predictive factors of CTLA-4 inhibitor-induced colitis, with the most evidence arising from studies on patients with advanced or metastatic malignant melanoma who received ipilimumab. Higher CD4^+^ T-cell and lower regulatory T-cell percentages were associated with colitis irAEs [112]. In addition, higher IL17 levels and sCTLA-4 levels greater than 200 pg/mL were also associated with colitis from ipilimumab, while higher IL6, IL8, and soluble CD25 levels were protective factors [69,70,112]. A study by Shahabi et al. focusing on serum RNA expression pre- and post-treatment showed that patients with GI irAEs had pretreatment overexpressions of multiple genes associated with immunity, cell cycle and proliferation, and intracellular vesicle trafficking compared to those without irAEs. Interestingly, the expressions of CD177 and CEACAM1, which are cell surface proteins on neutrophils, were markedly elevated three weeks post treatment in those with GI irAEs, but not pretreatment. However, this was noted to have high specificity but low sensitivity for predicting irAEs. The overexpression of immunoglobulin-related genes such as *IGHA1*, *IGHA2*, *IGHG1*, and *IGHV4-31* were also associated with GI irAEs [113].

In a study on 191 patients with solid tumors who received anti-PD(L)1 monotherapy, antinuclear antibody was associated with an increased risk of colitis irAEs [114]. Wölffer et al. revealed that increased white blood cell and absolute neutrophil counts were risk factors for patients with malignant melanoma. Interestingly, HLA-A homozygosity was also found to be associated with colitis; however, they did not elucidate if a specific HLA-A allele carried the elevated risk [114]. In another study, HLA-DQB1*03:01 was associated with colitis in malignant melanoma and NSCLC (OR 3.94, *p* = 0.017) [111].

##### Hepatobiliary and Pancreatic Dysfunction

An elevated white blood cell count was associated with increased risks of hepatitis and pancreatitis, while the absolute neutrophil count was associated only with pancreatitis [114]. A higher neutrophil–lymphocyte ratio was associated with reduced hepatitis irAE, as reported by Madjar et al. [100]. HLA-A homozygosity was again shown to be associated with pancreatitis, with another study by Jiang et al. revealing that HLA-A*26:01 was a risk allele for hyperbilirubinemia. In addition, *SMAD3* small sequence variations were also a risk factor for pancreatitis [114,116]. It is worth noting that autoimmune hepatitis antibodies (e.g., antinuclear antibody, anti-mitochondrial antibody, anti-LKM1, etc.) were not associated with the development of hepatitis irAEs [115].

#### 2.2.4. Rheumatologic

Surprisingly, no studies have reported the association between antinuclear antibody and the risk of rheumatologic irAEs. Only one study by Mathias et al. showed that pre-existing rheumatoid factor was associated with rheumatologic irAEs [117]. HLA-B*35:01 was reported to be a risk factor for any rheumatologic irAE, while HLA-DRB1*15:01 was a protective factor [29,34]. 

Cappelli et al. evaluated HLA allele subtypes in patients who received anti-PD(L)1-based therapy and observed an increased risk of inflammatory arthritis irAEs in patients with HLA-DRB1*04:05 (OR 8.6; 95% CI 1.7–43.4; *p* = 0.04) compared to the general population. Some trends toward increased risk in patients with HLA-B*52:01 and HLA-C*12:02 and decreased incidence in those with HLA-DQB1*03:01 were also observed. Interestingly, when compared to patients with rheumatoid arthritis, patients with ICI-induced inflammatory arthritis had a significantly lower prevalence of positive rheumatoid factor or anti-cyclic citrullinated protein [118]. No association between rheumatoid factor and irAEs was found in patients who received ipilimumab [15].

#### 2.2.5. Pulmonary

A study on 300 patients with NSCLC showed that an absolute eosinophil count of 125 or greater was associated with increased pneumonitis from ICI (adjusted OR 3.52; 95% CI 1.85–6.69; *p* < 0.001), while another study revealed increased white blood cell and decreased absolute lymphocyte counts as risk factors [119,120]. Of note, the association between the neutrophil–lymphocyte ratio and pneumonitis is still contradictory [100,120].

A study using an interferon-gamma (IFN-γ) release assay showed that NSCLC patients with low IFN-γ values of less than 10 or decreased IFN-γ levels by 3-6 weeks post treatment had more frequent pneumonitis from PD(L)1 inhibitors [121]. In two studies, elevated pretreatment levels of anti-CD74 and IL17 were found to be associated with pneumonitis in patients with solid tumors who received anti-PD1 with or without anti-CTLA-4 ICIs. However, another study by Miura et al. did not find this association [64,106,109].

Other biomarkers that have been reported to be associated with pneumonitis included HLA-B35, especially HLA-B*35:01, HLA-DRB1*11, and pre-existing anti-glutamic acid decarboxylase (anti-GAD) [23,34,122]. Interestingly, in a multivariate analysis, PD-L1 TPS was not significantly associated with pneumonitis in patients with NSCLC who received anti-PD(L)1 monotherapy [28].

#### 2.2.6. Other Organ Systems

Besides the organ systems mentioned above, irAEs can involve other uncommon sites. Möhn et al. utilized a neuroinflammatory cytokine/chemokine panel to screen for potential biomarkers for neurological irAE prediction. They found that MCP-1 and BDNF levels are positively associated with an increased risk of any grade ≥3 neurological irAE [123]. Another study on 290 patients with solid tumors who received atezolizumab reported a relationship between autoimmune encephalitis irAEs and HLA-B*27:05 [124]. Muller-Jensen et al. showed that positive neuromuscular autoantibodies were associated with ICI-induced neuromuscular disease, while brain-reactive autoantibodies were not predictive for autoimmune encephalitis (*p* < 0.0001) [125]. A reduced absolute lymphocyte count and elevated neutrophil–lymphocyte ratio were shown to be risk factors for myocarditis irAEs in patients with solid tumors who received ICI therapy [126]. Other HLA alleles that were reported to be associated with organ-specific irAEs include HLA-DPB1*04:02 and anemia and leukopenia, HLA-DRB3*01:01 and thrombocytopenia [116], and HLA-DRB1*04:05 and Vogt–Koyanagi–Harada-like uveitis [127]. 

## 3. Future Directions and Conclusions

Predictive biomarkers for irAE are one of the most important unanswered questions in the field of immuno-oncology. Unfortunately, at present, there are no effective biomarkers for predicting irAEs except TSH due to the inexpensive cost and widespread availability of the test. Periodic measurements of TSH before and during ICI therapy have been recommended by several screening guidelines, such as the American Thyroid Association and the European Society for Medical Oncology [130]. Although clinically available, antithyroid hormones are not routinely screened unless the patient has a previous diagnosis of autoimmune thyroid disease. Therefore, the clinical utility of this as a screening tool for thyroid irAEs remains uncertain and needs to be explored further for the accuracy and cost-effectiveness of the tests.

Other biomarkers are not yet at a stage to be successfully deployed into routine clinical practice due to uncertainties and variations in their predictive values. Specifically, a single biomarker is likely inadequate due to variations in the biology of the tumor and the immune pathways activated by each ICI. Moreover, several biomarkers suffer from non-unified cutoff values, rendering clinical usage and validation difficult. This resulted in the paucity of efforts and evidence required to reiterate the clinical utility of irAEs in general. We propose that predictive composite scores utilizing the clinical, pathologic, genetic, and gene/protein expression data might serve as a better alternative for stratifying irAE risk. Multiple risk scores have been created to support this need, as presented in Table 4.

Khan et al. developed psoriasis-associated polygenic risk scores from Immunochip and the UK Biobank to predict dermatologic irAEs, and the scores were then confirmed with whole-genome sequencing data from 479 patients with bladder cancer in the IMvigor211 trial. Higher psoriasis-associated polygenic risk scores were associated with a higher risk of dermatologic irAEs [133]. In another 140-gene germline polygenic risk score developed from a genome-wide association study (GWAS) on autoimmune thyroid disease from the UK Biobank, a model was trained using patients with solid tumors who received atezolizumab and was able to predict thyroid irAEs [134]. Using a deep neural network learning model, Sung et al. proposed a different set of 859-gene germline variants, HLA alleles, and blood counts. They could predict multiple organ-specific irAEs, including endocrine, pulmonary, and dermatologic irAEs [34].

Composite scores using RNA or cytokines have also been explored by several groups. An approach by Friedlander et al. using a 16-gene RNA signature to predict anti-CTLA-4-related colitis from tremelimumab was successfully performed with patients with advanced malignant melanoma [135]. Lim et al. studied a panel of 65 circulating cytokines with patients with malignant melanoma who received combined anti-PD1/anti-CTLA-4 therapy. They identified 11 cytokines (G-CSF, GM-CSF, fractalkine, FGF-2, IFNα2, IL1A, IL1B, IL1RA, IL2, IL12p70, and IL13) that were differentially upregulated in patients with severe irAEs both pre-treatment and 1-6 weeks post treatment in patients with severe irAEs and constructed a combined CYTOX score based on these cytokines. An elevated CYTOX score was shown to be associated with severe irAEs in both discovery and validation cohorts [136].

Interestingly, several drugs co-administered with ICIs have been shown to be associated with the increased risk of irAE. A pharmacovigilance study revealed a significant association between antibiotic use during ICI therapy and the development of irAEs in patients with lung cancer (OR 1.39, 95% CI 1.21–1.59 from FAERS and OR 1.32, 95% CI 1.09–1.59 from VigiBase databases) and pancreatic cancer (OR 4.61, 95% CI 2.56–8.32 from FAERS database) [137]. Chronic proton pump inhibitor use was also found to be adversely associated with GI irAEs in a cohort of 363 patients who received ICI (HR 13.22, 95% CI 3.11–56.10) [138]. Exposure to metformin, a drug used for type 2 diabetes mellitus, was recently shown to be associated with an increased risk of death compared to other non-insulin hypoglycemic agents despite its presumed antitumor and immunomodulating effects [139,140,141]. Further studies are needed to elucidate the causality of metformin and observed mortality, and whether this is associated with irAEs.

There still exist significant barriers for researching irAEs and the implementation of the biomarkers in clinical practice. First, the composite scores must be externally validated in other clinical settings before being implemented into widespread use. Second, one must make the complex tests more accessible in real-world settings. In other words, further research is necessary to create a practical tool for irAE prediction. For example, Zhukova et al. created a three-point scoring system consisting of the presence of HLA-DRB1, PD-L1 negativity, and dual ICI therapy and claimed its usefulness as a predictor for severe irAEs [132]. Third, most irAE studies have been performed in a retrospective fashion, requiring the vigilance of patients and physicians to document irAEs in patient charts. Therefore, irAEs can be underdiagnosed, jeopardizing the studies. Multiple approaches can be utilized to mitigate these limitations. Designing prospective studies that explore irAEs in a systematic screening and detection should be pursued. Patient-reported outcomes are another appealing option for incorporation in prospective studies in the future. An application-based platform for patients to directly report symptoms and communicate with the treatment team has been shown by Moradian et al. to be successful in this scenario as well [142]. Electronic patient-reported outcomes were also shown to improve the quality of life and reduce irAEs in patients who received ICIs [143]. Using likelihood scores that can detect irAEs from adverse events unrelated to ICIs, these types reported outcomes can also be very helpful, with such tools being already developed and confirmed in a cohort of 208 patients [131]. This may not be accomplished easily without extensive support and funding, and we believe that a large-scale collaboration or a cooperative oncology group can be a valuable resource for initiating such studies and validating on a larger population. Lastly, the studies discussed here were conducted prior to the approvals of other ICIs, such as anti-LAG3, for clinical use. Further work is needed to elucidate the mechanisms and potential biomarkers of irAE prediction for novel ICIs.

## Figures and Tables

**Figure 1 cancers-16-01225-f001:**
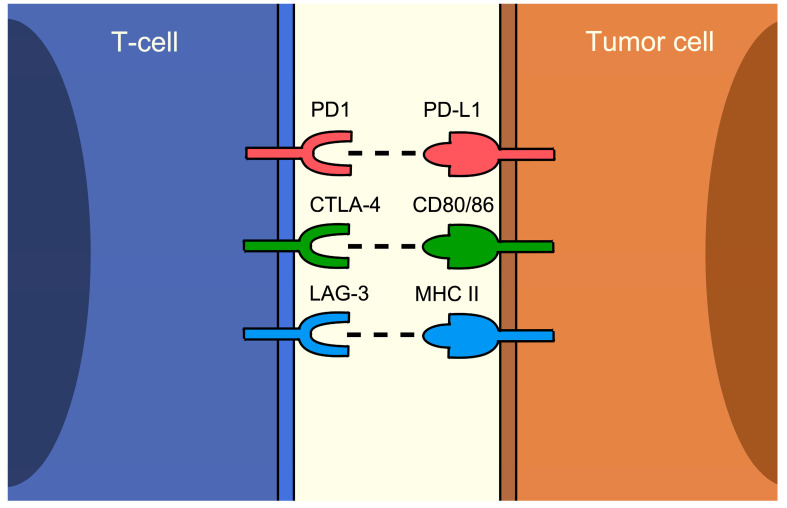
Immune checkpoint receptors and associated ligands on T- and tumor cells.

**Table 1 cancers-16-01225-t001:** Clinical abnormalities by organ system and type of ICI for any grade irAE.

Organ System	Frequency (%)
PD(L)1-inhibitors	
Colitis	<1–3
Pulmonary	<1–5
Skin	<1–16
Neurological	0.3
Endocrinopathy	7–23
Hepatic	<1–10
Renal	0–2
CTLA-4 inhibitors	
Colitis	7–15
Skin	19–34
Neurological	<1–4
Endocrinopathy	7–37
Hepatic	3–24
PD(L)1-inhibitors with CTLA-4 inhibitors	
Colitis	1–13
Pulmonary	3–7
Skin	16–30
Endocrinopathy	12–34
Hepatic	3–33
Renal	<1–7
PD1-inhibitors with LAG-3 inhibitors (from RELATIVITY-047 trial)	
Colitis	7
Pulmonary	4
Skin	9
Endocrinopathy	31
Hepatic	5
Renal	2

Abbreviation: CTLA-4, cytotoxic T-lymphocyte associated protein 4; ICI, immune checkpoint inhibitor; irAE, immune-related adverse event; LAG-3, lymphocyte-activation gene 3; PD(L)1, programmed cell death protein 1/programmed death-ligand 1.

**Table 4 cancers-16-01225-t004:** Predictive scoring systems for irAEs.

Scoring System	Association
6-item clinical likelihood score [131]	Score > 5: associated with higher risk of irAE
3-item clinicopathological score [132]	Score > 2: associated with severe irAE
Psoriasis-associated polygenic risk scores for dermatologic irAEs [133]	Higher score: associated with higher risk of dermatologic irAE
140-gene germline polygenic risk score [134]	Higher score: associated with thyroid irAE
859-gene germline variant, HLA alleles, and blood counts, deep neural network model [34]	Predictive of each organ-specific irAE
16-gene RNA expression signature [135]	Higher score at 30 days after treatment: increased risk of colitis
Combined 11 cytokine (CYTOX) score [136]	Higher score: associated with higher risk of any irAE

Abbreviation: irAE, immune-related adverse event.

## Data Availability

No new data were created or analyzed in this study. Data sharing is not applicable to this article.

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
