# Peer review of "Non-Invasive Predictive Biomarkers for Immune-Related Adverse Events Due to Immune Checkpoint Inhibitors"

_cancers, 2024, doi:10.3390/cancers16061225_

Round 1
Reviewer 1 Report
Comments and Suggestions for Authors
Altogether, I think that this manuscript is a useful contribution on immune-related Adverse Events associated with ICI therapy. There are some points, however, which, in my eyes, must be improved in order to make the manuscript acceptable for publication.
First of all, the authors should add a section on irAEs themselves: which are the most commonly observed that associate with one or the other ICI therapy etc.? I am well aware that there are many articles published over the years that deal with this topic but it would be, nevertheless, useful for the reader to have a bird’s eye view of these events in the introductory part of the manuscript. By the way, the manuscript, as it stands now, is not too long and such a section can be easily accommodated.
Second, the authors state that, currently, only TSH and thyroid autoantibodies are used as predictive biomarkers for ICI therapy. Could the authors enter a bit more in detail into this issue, e.g. which is their actual predictivity, how widespread is their use, when they are used a.s.o. This can be done in the “Conclusions” to the manuscript. Overall, the manuscript seems to me to very “dry”. I would suggest to make it a bit more colloquial.
Third, the are too many acronyms in the text. The authors should write the most important in extenso and perhaps skip the less relevant ones which can be included in the tables for the sake of completeness. Acronyms of genes should be written in italics.
Fourth, I miss some key references on the topic: e.g. Med. 2023;4:113-129; Oncoimmunology 2023;12:2204754; Crit Rev Immunol. 2022;42:1-9; Oncologist. 2024;29:e266-e274; Cancer Inform. 2023;22:11769351231178587.
Comments on the Quality of English LanguageOverall, the English of the manuscript is fine for me. There are some minor typos that can be amended after reviewing
Reviewer 2 Report
Comments and Suggestions for Authors
Ponvilawan B. et al. reviwed the Non-Invasive Predictive Biomarkers For Immune-related Adverse Events Due To Immune Checkpoint Inhibitors.
Given the increasing interets and use of ICI in cancers, the tpoic is well selected by teh authrs and it provides a good review about the adverse effects caused by ICI. However, it would be better to provide a definition of irAEs at the beginning. The readers need to know not only the definition bt also some examples of these adverse effects and their clinical manifestations. The authors address the point that irAEs might have been underdiagnosed in retrospective studies but do not provide a guideline or suggestion on how to address this problem bettter in prospective studies.
Reviewer 3 Report
Comments and Suggestions for Authors
The review entitled with “Non-Invasive Predictive Biomarkers For Immune-related Ad-2 verse Events Due To Immune Checkpoint Inhibitors” focused on the critical issue of irAEs associated with ICIs in cancer treatment, provides a more comprehensive overview of non-invasive predictive biomarkers during ICIs treatment. Overall, this review is well-structured and informative, but there is still some concerns. By addressing these points, this review can be even more valuable in clinical ICIs treatment.
Major concerns:
11. The introduction effectively addresses the importance of ICIs in cancer treatment and the challenges of irAEs, but if the authors can provide a brief overview of how ICIs works can strengthen the background information of the article.
22. In Line 59, the authors divided biomarkers into organ-nonspecific and organ-specific, they should briefly explain why this classification is relevant and how it aids in understanding and predicting irAEs.
33. While the review provides a comprehensive overview of potential biomarkers during ICIs treatment, it could benefit from a more robust discussion on the clinical implications of these findings. At this point, the authors should explain how could the identification of predictive biomarkers improve patient care and treatment outcome. As well, description about that if there are any ongoing efforts to validate these biomarkers in clinical practice is also important.
Minor concerns:
11. There are some typos:
a. In Line 310, "evaulated" should be "evaluated" .
b. In Line 471, "ICIs in for" should be "ICIs for"
22. In Line 288, There seems to be a redundant formatting instruction at the end of this paragraph.
Round 2
Reviewer 1 Report
Comments and Suggestions for Authors
The manuscript has been improved. There are still some typos that need to be amended. This is particularly true for the new parts that have been included. For this reason I enclose a version of the manuscript with my personal corrections. This should be of help in irder to save time.

The manuscript has been improved. There are still some typos that need to be amended. This is particularly true for the new parts that have been included. For this reason I enclose a version of the manuscript with my personal corrections. This should be of help in irder to save time.
